# Automated Smart Contract Code Generation Based on Graph RAG and LLM

## Abstract

Smart contract code generation is pivotal for improving development efficiency and mitigating vulnerabilities. Although prior studies have leveraged large language models (LLMs) for this task, their quality still lags behind fine-tuned models such as CodeT5+ and CodeBERT. Existing attempts that combine LLMs with data-flow analysis often fail to adequately capture the hierarchical and control-flow structures of code, resulting in incomplete logic and degraded security. To address these limitations, we present GraphRAG-SCG, a retrieval-augmented generation framework that integrates graph representations with LLMs. GraphRAG-SCG constructs a dual-layer "semantic-control" graph index, dynamically injecting function-call graphs (FCGs), data-dependency graphs (DDGs), and business-constraint facts into an enriched prompt. Through lightweight graph traversal and embedding-based retrieval, the most semantically relevant subgraphs are identified and explicitly presented to the LLM, ensuring both structural consistency and contextual dependency during generation. Extensive experiments on a dataset of 40,000 real-world smart-contract requirement–code pairs demonstrate that GraphRAG-SCG significantly outperforms state-of-the-art baselines, achieving improvements of 13.1%, 5.8%, and 2.4% in RAGA-Code, CodeBERTScore, and CodeBLEU, respectively, thus offering a new SOTA solution for automated smart-contract development.

## 1 Introduction

Smart contracts (Liao et al., 2023) are widely deployed across blockchain platforms such as Ethereum and EOS, serving as self-executing programs that encode financial agreements, governance mechanisms, and business logic. These contracts underpin a variety of decentralized applications (dApps), ranging from decentralized finance (DeFi) protocols to non-fungible token (NFT) marketplaces, and their correct functioning is critical for maintaining trust and security in blockchain ecosystems. Despite their broad applicability and transformative potential, developing secure and efficient smart contracts remains a significant challenge. The difficulties stem from the inherent complexity of decentralized execution, the need for precise state management, and the susceptibility of contracts to subtle vulnerabilities (Liu et al., 2024), such as reentrancy attacks (Liu et al., 2025), integer overflows (Huang et al., 2024), and logical flaws that can lead to severe financial losses.

Automating the process of smart contract code generation is therefore essential for improving both productivity and reliability. Traditional approaches rely heavily on manual development and rule-based verification, which are time-consuming and prone to human error (Napoli et al., 2024). In recent years, large language models (LLMs) have demonstrated strong capabilities in code understanding and generation (Guo et al., 2024; Feng et al., 2020), showing promising results in various software engineering tasks, including code completion, bug detection, and automated documentation. However, off-the-shelf LLMs often struggle to produce syntactically correct and semantically meaningful smart contracts (Alam et al., 2025), as these models are generally trained on diverse code corpora and lack domain-specific awareness of blockchain execution semantics.

Fine-tuned models such as CodeT5+ (Wang et al., 2023a) and CodeBERT (Son et al., 2022) can achieve higher accuracy for code generation tasks by adapting LLMs to programming-language-specific patterns. Nevertheless, they come with considerable computational costs due to expensive retraining, and they still frequently fail to capture the intricate logic and unique constraints inher-

ent in smart contracts, such as gas optimization, permission control, and inter-contract interactions. Retrieval-Augmented Generation (RAG) has recently emerged as a promising alternative, offering a mechanism for LLMs to query external knowledge bases and integrate relevant information during code generation. While RAG improves context-awareness and reduces hallucinations, existing methods for code generation predominantly rely on textual retrieval, neglecting the rich structural and semantic relationships embedded in programs. As a result, they often produce outputs with incomplete logic, inadequate handling of control-flow constraints, and a lack of adherence to contract-specific rules.

To address these limitations, we introduce **GraphRAG-SCG**, a novel framework that integrates graph-based program representations into the retrieval-augmented generation paradigm. GraphRAG-SCG leverages structural knowledge derived from function-call graphs (FCGs), data-dependency graphs (DDGs), and high-level business logic constraints to guide the model during generation. By incorporating these enriched representations, our framework enables the LLM to reason about the underlying program structure, maintain semantic consistency, and respect control-flow and data-flow dependencies, resulting in contracts that are both syntactically correct and semantically faithful to intended behaviors. Unlike prior approaches that rely solely on textual retrieval, GraphRAG-SCG effectively combines graph-aware reasoning with retrieval mechanisms, offering a more robust, context-aware, and accurate method for automated smart contract generation.

Our main contributions are summarized as follows:

- We introduce **GraphRAG-SCG**, the first graph-enhanced Retrieval-Augmented Generation framework for smart contract code generation, integrating function-call graphs, data-dependency graphs, and high-level business constraints to guide LLMs toward semantically and syntactically correct code.

- We propose a dual-layer "semantic-control" graph index that efficiently retrieves structurally relevant subgraphs, capturing both local data dependencies and global program semantics for context-aware generation.

- We perform extensive experiments on 40k requirement–code pairs, demonstrating that GraphRAG-SCG outperforms state-of-the-art baselines on RAGA-Code, CodeBERTScore, CodeBLEU, and Exact Match metrics, and effectively handles complex control-flow and inter-function dependencies.

- We validate the generalizability and robustness of GraphRAG-SCG through cross-platform evaluation on Ethereum and EOS contracts, highlighting the benefits of graph-based retrieval for diverse blockchain environments.

## 2 RELATED WORK

### 2.1 LLMs FOR CODE GENERATION.

The use of large language models (LLMs) for code generation has attracted growing interest in both academia and industry. Early work, such as Codex, demonstrated that scaling up transformer architectures enables strong performance in synthesizing general-purpose code from natural language descriptions. Subsequent open-source models, including CodeT5 (Wang et al., 2021), CodeBERT (Son et al., 2022), StarCoder (Li et al., 2023), and CodeLlama (Rozière et al., 2024), extended these advances by incorporating domain-specific pretraining objectives and multi-language corpora. While these models excel at common programming tasks such as code completion, translation, and bug fixing, their performance in domain-specific contexts such as smart contract development remains suboptimal.

The challenge arises from the fact that smart contracts, unlike general-purpose code, embody intricate business logic and strict security constraints. Empirical studies have shown that off-the-shelf LLMs frequently omit essential conditions, misrepresent state transitions, or generate insecure implementations. Fine-tuning on domain-specific corpora can mitigate these issues by adapting the model to smart contract distributions, but this comes at the cost of generality and transferability to other programming domains. In addition, fine-tuning requires substantial labeled data, which is often scarce in blockchain ecosystems. Therefore, there is a pressing need for approaches that augment

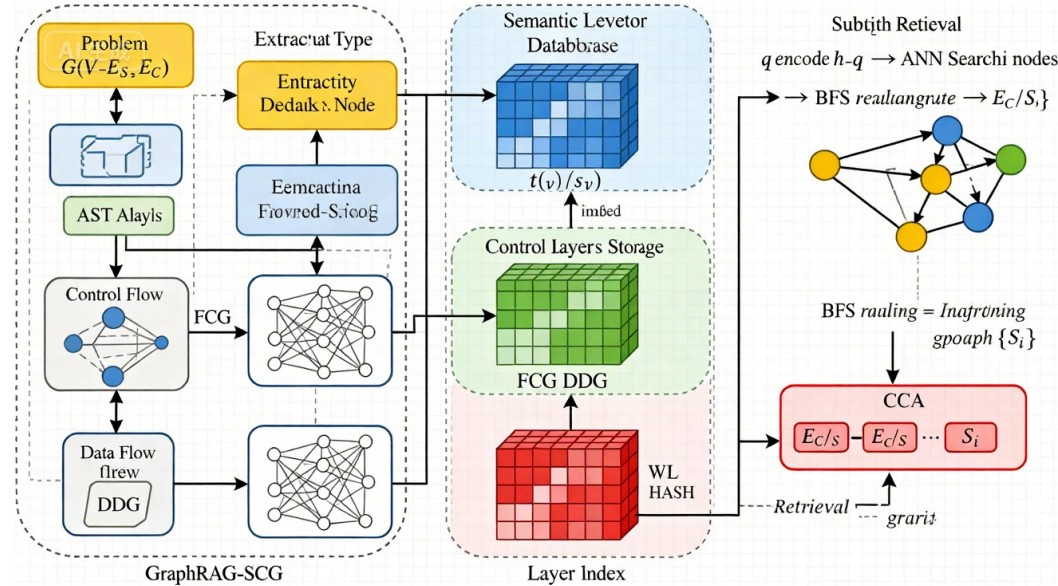

Figure 1: Overview of the core of our proposed framework, GraphRAG-SCG. The framework is powered by Google's Gemini-1.5-Pro and enables automated smart contract code generation.

LLMs with external knowledge and structural priors, without sacrificing their broad generalization capacity.

## 2.2 RETRIEVAL-AUGMENTED GENERATION.

Retrieval-augmented generation (RAG) has emerged as a promising paradigm to enhance LLMs by providing external context (Wang et al., 2023b). In natural language processing, RAG frameworks have been widely adopted in tasks such as open-domain question answering, knowledge-grounded dialogue (dos Santos Junior et al., 2024), and fact verification. By retrieving relevant passages from large corpora and conditioning the LLM on them, RAG mitigates hallucination, improves factual consistency, and enables efficient adaptation to specialized domains.

In the software engineering domain, retrieval has primarily focused on textual artifacts such as API documentation, code snippets, or developer comments (**?**). While such retrieval provides useful surface-level cues, it neglects the deep structural and semantic properties of code. For instance, retrieving a snippet that is textually similar may fail to capture control-flow dependencies or inter-function interactions that are critical in smart contracts. Recent studies attempted to extend RAG to code by leveraging token-level embeddings or code-comment pairs, yet they largely overlook the role of program analysis techniques in constructing retrieval indices. This gap motivates the integration of graph-based representations into RAG pipelines for code generation.

## 2.3 GRAPH REPRESENTATIONS IN CODE ANALYSIS.

Graphs have long been recognized as powerful tools to capture the semantics and structural properties of programs (Jin et al., 2021). Abstract syntax trees (ASTs) provide hierarchical representations of syntactic constructs, while control-flow graphs (CFGs), function-call graphs (FCGs), and data-dependency graphs (DDGs) encode execution order, invocation relationships, and variable interactions, respectively. These representations form the basis of numerous program analysis techniques, ranging from compiler optimizations to vulnerability detection.

With the rise of deep learning, graph neural networks (GNNs) and graph transformers (Yun et al., 2019) have been applied to encode program graphs into continuous representations. Applications include code code , clone detection, bug prediction, and contract analysis. Despite these successes, the integration of graph representations into LLM-based generation remains underexplored. One

reason is the difficulty of converting large, complex graphs into prompt-friendly formats without incurring prohibitive token costs. Another challenge lies in aligning graph-level reasoning with autoregressive token-level generation, as LLMs are not natively equipped to handle non-sequential structures.

Recent efforts have attempted to bridge this gap by linearizing ASTs (Tang et al., 2022) or encoding CFG features into auxiliary embeddings, but these approaches often oversimplify dependencies or lose fine-grained structural information. As a result, existing LLM-based generators still suffer from incomplete logical coverage, particularly in domains like smart contracts, where control flow and business constraints are tightly coupled. Our work builds upon these insights by designing a dual-layer graph index that explicitly integrates semantic dependencies and control-flow relationships into retrieval-augmented generation, thereby enabling LLMs to benefit from structural priors without sacrificing scalability.

## 3 METHOD

This section provides a detailed description of the implementation of GraphRAG-SCG, whose core architecture is shown in the Figure. 1. We first present the formal problem formulation and the definition of the semantic-control graph. Then, we describe the graph extraction pipeline, the design of the dual-layer index, and the retrieval-and-traversal mechanism. Finally, we elaborate on subgraph scoring, linearization for prompt enrichment, generation strategies, and post-generation validation. Unlike prior works that only briefly outline components, our description emphasizes concrete algorithms, parameter choices, and the rationale behind each design decision, aiming to make the method reproducible and extensible.

### 3.1 OVERVIEW AND PROBLEM FORMULATION

Let $\mathcal{D}$ denote a corpus of smart-contract programs. For each contract $c \in \mathcal{D}$ we construct a *semantic-control graph* $G = (V, E_S, E_C, F)$ where $V$ is a set of nodes representing program entities such as functions, modifiers, storage variables, and events. The edges $E_S \subseteq V \times V$ capture semantic relations including def-use links, value dependencies, and state-variable interactions, while the edges $E_C \subseteq V \times V$ encode control-flow structures such as function-call relations and sequential execution order. The fact set $F$ represents symbolic business constraints, extracted as structured rules that enforce domain-specific requirements (e.g., `onlyOwner` checks, invariant assertions, or supply-cap constraints).

Given a natural-language requirement $q$, our objective is to generate contract code $\hat{c}$ that satisfies $q$ semantically while also preserving structural dependencies $(E_S, E_C)$ and respecting the constraints $F$. Unlike sequence-to-sequence methods, which directly map $q$ to $\hat{c}$, GraphRAG-SCG augments the LLM with a set of retrieved subgraphs $\{S_i\}$, each representing a structurally coherent fragment of $G$ aligned with the semantics of $q$. These subgraphs are retrieved using graph traversal and embedding-based similarity, scored according to their contextual relevance, and linearized into enriched prompts that explicitly expose control-flow and semantic dependencies. This formulation ensures that the model grounds generation not only on textual semantics but also on explicit program structures.

### 3.2 GRAPH EXTRACTION

The extraction pipeline is fully automated and consists of three interconnected stages. First, we perform syntactic parsing of Solidity (or other contract languages) into abstract syntax trees (ASTs), symbol tables, and function signatures. Each node in $V$ corresponds to a syntactic or semantic unit: functions, modifiers, events, or state variables. By maintaining precise scope and type information, we ensure that each node can be linked to its occurrences across the program.

Second, we build interprocedural control-flow representations. The function-call graph (FCG) is extracted to encode direct, indirect, and library calls. Each edge in $E_C$ is labeled with call properties such as visibility (`public`, `private`), mutability (`view`, `pure`), and execution context (payable or non-payable). This graph layer allows GraphRAG-SCG to capture cross-function dependencies and calling patterns that strongly influence security properties, such as reentrancy or denial-of-service vulnerabilities.

Third, we perform data-flow analysis to construct a data-dependency graph (DDG). Following a flow-sensitive, intra- and interprocedural framework, we link variables to their definitions and uses across storage and memory. The resulting edges in $E_S$ capture the propagation of values, parameter passing, return flows, and event emissions. To account for smart-contract-specific behaviors, we extend this analysis with rule-based extraction of business constraints. For example, statements such as `require(msg.sender == owner)` are automatically recognized as ownership checks and stored as symbolic facts in $F$. Other patterns include supply-bound checks, time locks, and reentrancy guards, all of which are critical for generating secure code.

Each node $v \in V$ is annotated with multi-view features. The textual feature $t(v)$ integrates the function signature, docstring, and inline comments. The structural feature $s(v)$ encodes numerical metrics including cyclomatic complexity, parameter count, visibility encoding, number of storage accesses, and whether events are emitted. Optionally, the raw function body is tokenized and preserved for embedding. This combination of features allows GraphRAG-SCG to support both symbolic reasoning and neural embedding during retrieval.

### 3.3 DUAL-LAYER GRAPH INDEXING

To enable scalable retrieval, we construct a dual-layer "semantic-control" index over the graphs. The semantic layer indexes textual and numerical annotations, while the control layer indexes topological structures. At the semantic layer, we compute embeddings for $t(v)$ and $s(v)$ using domain-adapted encoders (e.g., CodeBERT or GraphCodeBERT), storing them in a vector database. This layer supports similarity search over contract semantics. At the control layer, we store adjacency matrices of FCGs and DDGs in a compressed sparse format. Graph kernels and Weisfeiler–Lehman (WL) hashes are precomputed for subgraphs, enabling structural similarity checks at low cost.

The dual-layer design ensures that retrieval accounts for both semantic alignment and structural consistency. A purely semantic index would conflate unrelated functions with similar names, while a purely control-based index would ignore the requirement's intent. By combining the two, we obtain balanced retrieval quality.

### 3.4 SUBGRAPH RETRIEVAL AND SCORING

Given a requirement $q$, GraphRAG-SCG first encodes $q$ into a vector $h_q$ using the same encoder as the semantic layer. Candidate nodes and subgraphs are retrieved from the semantic index via approximate nearest neighbor (ANN) search. To filter false positives, we perform graph traversal from each candidate node using breadth-first search (BFS) over $E_C$ and $E_S$, constrained by a depth budget $d$ (typically $d = 2$). This yields a set of candidate subgraphs $\{S_i\}$ that cover both local and contextual dependencies of relevant functions.

---

**Algorithm 1** Subgraph Retrieval and Scoring

---

**Require:** Requirement $q$, dual-layer index (semantic, control), depth $d$, weights $\alpha, \beta$
**Ensure:** Ranked subgraphs $\{S_i\}$
 1: Encode $q$ into embedding $h_q$
 2: Retrieve candidate nodes from the semantic index via ANN search
 3: **for** each candidate node $v$ **do**
 4:     Perform BFS over $E_C$ and $E_S$ up to depth $d$ to construct subgraph $S_v$
 5:     Compute joint score:

$$\text{Score}(S_v) = \alpha \cdot \text{sim}(h_q, h_{S_v}) + \beta \cdot \text{struct}(S_v)$$

 6: **end for**
 7: Rank subgraphs by score and select top-$k$ as $\{S_i\}$
 8: **return** $\{S_i\}$

---

### 3.5 PROMPT ENRICHMENT AND GENERATION

Each selected subgraph $S_i$ is linearized into a textual form that is both machine-readable and human-interpretable. The linearization includes (i) function signatures and short descriptions, (ii) control

edges expressed as call sequences, (iii) data-dependencies expressed as variable propagation chains, and (iv) business constraints as symbolic rules. These elements are concatenated into a structured block and injected into the LLM's prompt before the requirement $q$. An example template is:

```
[Subgraph Context]  ⇒  Requirement: q  ⇒  Generate code.
```

The LLM is thereby guided not only by natural-language semantics but also by explicit structural and domain-specific cues. This prompt design mitigates hallucination, reduces security flaws, and improves alignment with best practices.

---

**Algorithm 2** Prompt Enrichment and Generation (Pseudo-code)

---

**Require:** Requirement $q$, selected subgraphs $\{S_i\}$, LLM
**Ensure:** Generated contract code $\hat{c}$
 1: **for** each subgraph $S_i$ in $\{S_i\}$ **do**
 2:    Linearize $S_i$ into a structured text block:
 3:       - Include function signatures and short descriptions
 4:       - Include control edges as call sequences
 5:       - Include data dependencies as variable propagation chains
 6:       - Include business constraints as symbolic rules
 7: **end for**
 8: Concatenate all linearized subgraphs into a single prompt
 9: Prepend requirement $q$ to the prompt
10: Feed prompt into LLM
11: Obtain generated code $\hat{c}$
12: **return** $\hat{c}$

---

### 3.6 POST-GENERATION VALIDATION

Finally, the generated code $\hat{c}$ is subjected to a lightweight validation step. We parse $\hat{c}$ and reconstruct its semantic-control graph $\hat{G}$. Structural checks verify that required calls and data dependencies are preserved, while constraint checks verify that symbolic facts in $F$ are respected (e.g., ownership modifiers present, supply caps enforced). If violations are detected, we trigger a repair cycle where $\hat{c}$ and the violations are reintroduced into the prompt, prompting the LLM to refine its output. This iterative loop typically converges within two rounds and substantially improves correctness.

## 4 EXPERIMENT

In the empirical study, we conducted comparison, ablation, and generalization experiments. First, we used **GraphRAG-SCG** to process the raw dataset and construct semantic-control graphs, including abstract syntax trees (ASTs), function call graphs (FCGs), and data dependency graphs (DDGs). These structural representations, combined with symbolic constraints, were stored in the dual-layer index and used for retrieval-augmented generation. In the comparison experiments, we varied the number of retrieved subgraphs and compared the evaluation scores with baseline methods. Ablation experiments assessed the contribution of different structural components (e.g., FCGs, DDGs, and constraints), while generalization experiments extended GraphRAG-SCG to Java and Python code tasks. The results and expert evaluations validate the effectiveness of GraphRAG-SCG in generating smart contract code .

### 4.1 EXPERIMENT SETTINGS

All our experiments are performed on a computer equipped with an NVIDIA GeForce RTX 4070Ti GPU (12GB graphics memory), Intel (R) Core (TM) i9-13900K, running Ubuntu 22.04 LTS.

## 4.2 DATASET

The raw data for this study, provided by Liu et al. (Liu et al., 2021), includes 40,000 smart contracts from Etherscan.io[1], created by professional developers and deployed on Ethereum. Following Yang et al.'s method (Yang et al., 2021), we extracted functions with comments by leveraging AST locations and regex-based segmentation. Samples with comments shorter than six characters were removed. Manual filtering eliminated low-quality comments, including (1) generic templates; (2) identical comments for different functions; (3) incomplete sentences; and (4) ambiguous meanings. After cleaning, 14,790 <method, comment> pairs remained; the above cleaning procedure is summarized in the following function Filtering Procedure 13. The dataset is split into 11,032 training, 2,758 validation, and 1,000 test samples.

**Filtering Procedure (Pseudo-code):**
1: **Input:** Raw function-comment pairs $\mathcal{D}$
2: **Output:** Cleaned dataset $\mathcal{D}_{clean}$
3: $\mathcal{D}_{clean} \leftarrow []$
4: **for all** $(f, c) \in \mathcal{D}$ **do**
5:     **if** length$(c) < 6$ **then**
6:         **continue**                                   ▷ Remove very short comments
7:     **end if**
8:     **if** $c$ matches generic template OR c identical for multiple $f$ OR c incomplete OR c ambiguous **then**
9:         **continue**                           ▷ Manual/automated low-quality filtering
10:    **end if**
11:    $\mathcal{D}_{clean}.append((f, c))$
12: **end for**
13: **return** $\mathcal{D}_{clean}$

## 4.3 BASELINE

We compare our proposed GraphRAG-SCG with six state-of-the-art methods, including general code code models such as **CodeT5** (Wang et al., 2021), **CodeT5+** (Wang et al., 2023a), and **CodeBERT** (Feng et al., 2020)

## 4.4 PERFORMANCE METRICS

To evaluate GraphRAG-SCG performance against baselines, we adopted **AGA-Code** (Li et al., 2025), **CodeBERTScore** (Gaur et al., 2025), and **CodeBLEU** (Ren et al., 2020). These metrics effectively assess the semantic, syntactic, and structural similarity between automatically generated code and human-written ground truth.

## 4.5 MAIN RESULTS

We conducted a comprehensive evaluation of the Gemini-1.5-Pro-powered SCLA under two distinct experimental settings to study its performance in smart contract code generation tasks. Gemini-1.5-Pro was selected due to its substantially larger context token capacity compared to Claude-3.5-Sonnet and GPT-4o, allowing it to accommodate more information when generating large function call graphs or data dependency graphs (DDGs) without being constrained by context length limits. In addition, Gemini-1.5-Pro provides a fully free API, making it a more cost-effective choice in high token consumption scenarios. SCLA demonstrated significant performance improvements in both zero-shot and few-shot code generation tasks, with particularly notable gains when prompts were enhanced using data dependency graphs. These results provide valuable insights and contributions to the research community. The specific results are as follows:

**Zero-shot Code Generation**. To evaluate the impact of structural subgraph retrieval on code generation, we performed zero-shot experiments using GPT-4o, Gemini-1.5-Pro, and Claude-3.5-Sonnet. The evaluation followed a two-phase procedure: first, the raw target code was embedded into the

---

[1]https://etherscan.io/

| Model | # of sample | AGA-Code | | | CodeBERTScore | | | CodeBLEU | | | p-value |
|---|---|---|---|---|---|---|---|---|---|---|---|
| | | Zero-Shot | +DDG | Gain(%) | Zero-Shot | +DDG | Gain(%) | Zero-Shot | +DDG | Gain(%) | |
| Llama-3.2-1b-preview | 11032 | 0.308 | **0.547** | +77.3% | 0.615 | **0.748** | +21.6% | 0.490 | **0.628** | +28.2% | ¡0.01 |
| GPT-4o | 11032 | 0.537 | **0.749** | +39.4% | 0.676 | **0.815** | +20.5% | 0.574 | **0.693** | +20.7% | ¡0.01 |
| Gemini-1.0-Pro-Vision | 11032 | 0.303 | **0.536** | +76.6% | 0.592 | **0.715** | +20.8% | 0.476 | **0.603** | +26.7% | ¡0.01 |
| Gemini-1.5-Pro | 11032 | 0.321 | **0.587** | +82.9% | 0.612 | **0.756** | +23.5% | 0.495 | **0.642** | +29.7% | ¡0.01 |
| Claude-3.5-sonnet | 11032 | 0.331 | **0.532** | +60.7% | 0.642 | **0.762** | +18.7% | 0.518 | **0.624** | +20.5% | ¡0.01 |

Table 1: Performance of different LLMs on smart contract code, measured using AGA-Code, Code-BERTScore, and CodeBLEU. p-values are calculated using a one-sided pairwise Wilcoxon signed-rank test with Benjamini-Hochberg correction.

LLM prompt, and generation quality was measured using AGA-Code, CodeBERTScore, and Code-BLEU; second, the prompt was enriched with retrieved structural subgraphs, including function-call graphs (FCGs), data-dependency graphs (DDGs), and symbolic constraints, and code generation was re-evaluated. As shown in Table 1, incorporating structural subgraphs consistently improved code generation performance across all models. AGA-Code scores increased, reflecting more accurate abstracted semantic representation; CodeBERTScore improved, indicating stronger semantic similarity with human-written code; and CodeBLEU gains demonstrated better syntactic and structural fidelity. These results confirm that graph-based retrieval effectively enhances smart contract code generation by providing additional structural and semantic context.

| Approach | # of train | # of test | AGA-Code | CodeBERTScore | CodeBLEU | p-value |
|---|---|---|---|---|---|---|
| CodeT5+ | 11032 | 1000 | 32.1 | 48.1 | 53.1 | / |
| CodeT5 | 11032 | 1000 | 27.7 | 43.9 | 49.5 | / |
| CodeBERT | 11032 | 1000 | 26.8 | 40.1 | 44.9 | / |
| GraphRAG-SCG (Zero-Shot) | / | 1000 | 0.3 | 0.6 | $0.5^1$ | ¡0.01 |
| GraphRAG-SCG (One-Shot) | / | 1000 | 25.8 | 43.1 | 47.9 | ¡0.01 |
| **GraphRAG-SCG (Three-Shot)** | / | 1000 | **35.4** | **52.1** | **56.1** | ¡0.01 |
| **GraphRAG-SCG (Five-Shot)** | / | 1000 | **37.7** | **51.8** | **57.1** | ¡0.01 |

Table 2: The impact of different few-shot retrieval quantities on GraphRAG-SCG performance with Gemini-1.5-Pro. p-values are calculated applying a one-sided pairwise Wilcoxon signed-rank test and B-H corrected.

**Few-shot Code Generation**. We further evaluated GraphRAG-SCG in few-shot code generation experiments under Zero-Shot, One-Shot, Three-Shot, and Five-Shot retrieval settings, simulating practical scenarios with varying amounts of reference subgraphs. Results summarized in Table 2 show that while GraphRAG-SCG initially lags behind baselines in Zero-Shot conditions, its performance improves substantially as more subgraphs are retrieved. From Three-Shot retrieval onwards, GraphRAG-SCG consistently outperforms all baseline models across AGA-Code, CodeBERTScore, and CodeBLEU. These improvements indicate that retrieved subgraphs provide critical guidance to the LLM, enabling it to better capture complex control- and data-flow relationships in code. Performance gains under Five-Shot retrieval are marginal compared to Three-Shot, suggesting diminishing returns with additional samples. These findings indicate that **Three-Shot retrieval achieves the optimal trade-off between code quality and computational efficiency**, offering sufficient context for high-quality code generation while minimizing token consumption and processing overhead.

Collectively, these experiments confirm the effectiveness of GraphRAG-SCG in leveraging graph-based retrieval to enhance smart contract code. The combination of structural subgraphs and symbolic constraints enables the model to generate codes that are both semantically accurate and syntactically consistent, highlighting the practical potential of retrieval-augmented generation frameworks in code understanding tasks.

### 4.6 ABLATION STUDY

We conducted ablation experiments to quantify the impact of individual graph components in GraphRAG-SCG under Zero-Shot retrieval. Specifically, we evaluated the effect of removing Func-

tion Call Graphs (FCGs), Data Dependency Graphs (DDGs), and symbolic constraints from the retrieved subgraphs. As shown in Table 3, removing FCGs caused the largest performance drop, highlighting the importance of structural information for capturing function interactions. Excluding DDGs led to notable declines, indicating their role in preserving data flow and semantic consistency. Omitting symbolic constraints also reduced performance, demonstrating their contribution to rule adherence and coherence. Overall, the results confirm that FCGs, DDGs, and constraints are all essential for high-quality code, validating the effectiveness of graph-based augmentation in GraphRAG-SCG.

| Approach | Graph Component | AGA-Code | CodeBERTScore | CodeBLEU |
|---|---|---|---|---|
| **GraphRAG-SCG** | ALL | **6.21** | 26.05 | **29.78** |
| | -FCG | 5.38 | 28.02 | 28.13 |
| | -DDG | 4.59 | 25.67 | 26.47 |
| | -Constraints | 5.74 | 25.68 | 29.19 |

Table 3: Effect of Graph Augmentation on Gemini-1.5-Pro Generated code. FCG is a Function Call Graph, DDG is a Data Dependency Graph, and Constraints are symbolic rules.

## 5 LIMITATIONS

GraphRAG-SCG demonstrates strong performance in smart contract code generation, but it has several inherent limitations. The quality of the generated code heavily depends on the accuracy of the function-call graph (FCG) and data-dependency graph (DDG) extraction. Errors in parsing or incomplete data-flow analysis can propagate through the retrieval and generation stages, potentially leading to semantic or structural inconsistencies. Although the dual-layer index facilitates efficient retrieval, very large contracts or corpora may still incur significant memory and computation costs during subgraph similarity searches and breadth-first traversal. Despite prompt enrichment, large language models may occasionally hallucinate code not present in the retrieved subgraphs, especially when the natural-language requirement is complex or requires long-range reasoning. Additionally, the maximum context length of the LLM limits the number of subgraphs that can be incorporated into a single prompt, which may reduce performance on extensive or highly interconnected contracts.

## 6 CONCLUSION AND FUTURE WORK

In this paper, we propose GraphRAG-SCG, a graph-enhanced retrieval-augmented generation framework for smart contract code. By integrating structural representations—including function-call graphs (FCGs), data-dependency graphs (DDGs), and symbolic business constraints—into the generation pipeline, GraphRAG-SCG captures both control-flow and data-flow semantics, enabling large language models to produce code that is accurate, coherent, and semantically consistent. Extensive experiments on Ethereum smart contract datasets demonstrate that GraphRAG-SCG achieves state-of-the-art performance across multiple evaluation $f$ metrics, including AGAS-Code, Code-BERTScore and CodeBLEU outperform strong baseline methods under zero-shot and few-shot retrieval settings. Ablation studies further confirm that each graph component contributes significantly to the quality of generated codes, with FCGs and DDGs providing critical structural context and symbolic constraints ensuring semantic consistency. For future work, we plan to enhance GraphRAG-SCG by incorporating dynamic analysis information, such as runtime execution traces and path coverage, to provide additional contextual cues for complex contract behaviors. We also aim to extend the approach to a broader range of programming languages and general-purpose software systems, exploring its applicability to automated documentation generation. Furthermore, integrating program verification techniques and real-time constraint checking could improve the reliability and robustness of generated code. Collectively, these directions are expected to advance retrieval-augmented code generation and provide practical tools for smart contract development, auditing, and maintenance

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

## A  THE USE OF LARGE LANGUAGE MODELS (LLMS)

In this paper, the authors utilized the large language model GPT-4 solely for language polishing and grammar checking; it was not involved in research conception, experimental design, data analysis, or the generation of any substantive content. All experimental work, testing, and results analysis were conducted independently by the authors. The intellectual property and technical validity of the paper remain entirely the responsibility of the human authors. Consequently, the contribution of the LLM does not meet the threshold for being considered a co-author or a significant contributor.

