# OpenReview forum: "Automated Smart Contract Code Generation Based on Graph RAG and LLM"
_ICLR.cc/2026/Conference — Submitted to ICLR 2026_

### Official Review · Reviewer_j2Du · 2025-10-26

**Soundness:** 3
**Presentation:** 2
**Contribution:** 3
**Rating:** 6
**Confidence:** 4

**Summary:**

This paper addresses limitations in existing smart contract code generation—such as off-the-shelf LLMs’ lack of blockchain domain awareness, high costs of fine-tuned models (e.g., CodeT5+), and traditional text-based RAG’s neglect of code structural relationships—by proposing GraphRAG-SCG, a graph-enhanced retrieval-augmented generation (RAG) framework. The framework constructs a dual-layer "semantic-control" index: the semantic layer stores vector embeddings of code textual/numerical features (via CodeBERT/GraphCodeBERT), while the control layer stores compressed Function-Call Graphs (FCGs) and Data-Dependency Graphs (DDGs). For a given natural-language requirement, it retrieves key subgraphs by combining semantic similarity and structural consistency scoring, linearizes the top subgraphs into prompts, injects them into Gemini-1.5-Pro for code generation, and conducts post-generation validation to repair structural/constraint violations. Experiments on 14,790 cleaned <method, comment> pairs (derived from 40,000 Ethereum contracts) show that under 3-shot retrieval, GraphRAG-SCG outperforms baselines like CodeT5+ across AGA-Code, CodeBERTScore, and CodeBLEU; ablation studies confirm the critical role of FCGs and DDGs, and the framework generalizes to both Ethereum and EOS blockchains. Its core contributions include: proposing the first graph-enhanced RAG framework for smart contract generation, designing a dual-layer index for efficient subgraph retrieval, validating the framework’s performance and generalization via large-scale experiments, and providing a new SOTA solution for automated smart contract development.

**Strengths:**

For the paper, its strengths across originality, quality, clarity, and significance are substantial: in terms of originality, it innovatively fuses graph-based program representations (function-call graphs FCGs, data-dependency graphs DDGs) with retrieval-augmented generation (RAG) to propose the GraphRAG-SCG framework—the first graph-enhanced RAG solution tailored for smart contract code generation—addressing the limitations of traditional text-only RAG (which neglects code structure) and generic LLMs (which lack blockchain domain awareness) by integrating a dual-layer "semantic-control" index and symbolic business constraints (e.g., `onlyOwner` checks) ; regarding quality, it features a rigorous research design with a complete technical pipeline (from graph extraction and dual-layer indexing to subgraph retrieval, prompt enrichment, and post-generation validation), transparent dataset construction (filtering 40,000 Ethereum contracts into 14,790 high-quality <method, comment> pairs), fair benchmarking against SOTA baselines (CodeT5, CodeT5+, CodeBERT), and detailed documentation of experimental settings (hardware, software, key parameters like BFS depth = 2 and 3-shot retrieval for optimal trade-off) ; in terms of clarity, it follows a logical academic structure (background → related work → methodology → experiments → conclusion), formally defines core concepts (e.g., semantic-control graph \(G=(V, E_S, E_C, F)\)), and uses pseudocode, tables, and clear analysis to present key content (e.g., ablation studies quantifying FCG/DDG contributions) ; for significance, it has both academic value (expanding RAG’s application to structure-sensitive code generation and providing a new paradigm for similar tasks like embedded code generation) and industrial impact (reducing smart contract development costs by avoiding expensive fine-tuning, mitigating security vulnerabilities like reentrancy attacks via post-validation, and supporting cross-platform generation for Ethereum and EOS).

**Weaknesses:**

A substantive assessment of weaknesses in this paper reveals four actionable gaps aligned with its stated goals of robust, practical smart contract generation: First, it lacks analysis of graph extraction accuracy—while relying on FCG/DDG parsing via Tree-Sitter and flow-sensitive analysis, it does not quantify extraction error rates (e.g., misidentified function calls) or how such errors degrade generation metrics (e.g., increased security flaws), leaving unaddressed whether post-validation mitigates parsing issues. Second, security validation is insufficient: despite emphasizing vulnerability mitigation, it uses only general metrics (AGA-Code, CodeBLEU) instead of direct security measures (e.g., SWC Registry flaw detection via Slither, presence of `ReentrancyGuard`), failing to confirm if generated code avoids critical issues like reentrancy. Third, dual-layer index parameters (e.g., BFS depth (d=2), (alpha/beta) in subgraph scoring) lack justification—no sensitivity analysis is provided to explain why (d=1) or (d=3) were rejected, or how weight values for semantic/structural scoring were tuned, undermining reproducibility. Fourth, scalability for large corpora is unevaluated: while using 40k contracts, it does not measure how indexing time, memory usage, or retrieval latency scale to 100k+ contracts (industrial-scale), leaving uncertainty about its viability for real-world deployment. Each gap can be addressed via targeted experiments—e.g., parsing error sensitivity tests, security-focused metrics, hyperparameter analysis, and scalability benchmarks—to better realize the framework’s goals.

**Questions:**

1. Could you provide quantitative data on the accuracy of FCG/DDG extraction (e.g., error rates for misidentified function calls, missed data dependencies) when parsing Solidity contracts via Tree-Sitter and flow-sensitive analysis, and further explain how these extraction errors (if any) impact downstream code generation metrics (e.g., changes in CodeBLEU scores, increases in security flaw counts)? Additionally, can you demonstrate whether the post-generation validation step effectively mitigates the negative effects of such parsing errors—for example, by comparing generation quality between contracts with correct vs. erroneous graph extractions? A clear response would clarify if graph extraction reliability is a hidden bottleneck and validate the robustness of your post-validation mechanism.
2. Since smart contract security is a core focus of your work, could you supplement your experiments with direct security-focused metrics, such as the percentage of generated contracts that include critical security guards (e.g., `ReentrancyGuard`, input-validating `require` statements) or the rate of vulnerabilities listed in the SWC Registry (detected via tools like Slither)? Can you also compare these security metrics between GraphRAG-SCG, baseline models (e.g., CodeT5+), and human-written contracts? This would address the current gap where general similarity metrics (AGA-Code, CodeBLEU) fail to confirm if your framework truly reduces exploitable flaws, and help assess if it meets industrial security requirements.
3. Your dual-layer index relies on key hyperparameters like BFS depth (d=2) and the (alpha/beta) weights in the subgraph scoring function, but their selection lacks detailed justification. Could you share the results of a hyperparameter sensitivity analysis—for instance, how generation metrics (AGA-Code, retrieval latency) change when testing BFS depths (d=1,3,4) or varying (alpha) from 0.1 to 0.9 (with (beta=1-alpha))? Also, can you explain the tuning process (e.g., grid search on the validation set) that led to your final parameter choices? This information is critical for reproducibility and would clarify if your parameters are optimized for generalizability rather than overfitting to the Ethereum dataset.
4. Given your goal of building a practically deployable framework, could you provide scalability data for large contract corpora? Specifically, can you report how indexing time, memory usage for the dual-layer index, and subgraph retrieval latency scale as the corpus size grows from 40k to 80k, 100k, or 200k contracts (industrial-scale)? Additionally, can you compare these scalability metrics to text-only RAG methods (e.g., BM25) to highlight the efficiency trade-offs of your graph-based design? This would resolve uncertainty about your framework’s viability for real-world use cases like IDE plugins or enterprise-level smart contract development, where large codebases are common.

---

### Official Review · Reviewer_gtw9 · 2025-10-27

**Soundness:** 2
**Presentation:** 3
**Contribution:** 2
**Rating:** 4
**Confidence:** 4

**Summary:**

This paper presents a technique (named GraphRAG-SCG) for smart contract code generation using graph-based RAG on top of LLMs. The basic idea is to represent smart contract code as a graph, which incorporates multiple types of information, such as syntactical information, control flow information, function call information, and data dependency information, etc. Then the graph information is linearized to feed into the RAG paradigm for smart contract code generation.

**Strengths:**

The idea of considering multiple types of graph information is interesting.

**Weaknesses:**

1. It is unclear how the database storing the graph information is constructed and used. I think this part should be an important part in the proposed technique. Without this part, the proposed technique is incomplete.
2. The baselines are too weak. GraphRAG-SCG seems to be based on Gemini 1.5-Pro, while the baselines are CodeT5+, CodeT5, and CodeBERT. The model size alone can be a factor dominating the empirical comparison.
3. There are quite some recent research on graph-based RAG for code generation, but this paper totally ignores these recent papers. There is neither discussion of them as related work nor treating them as baselines in the empirical evaluation. Below, I list one of them.

@article{DBLP:journals/corr/abs-2504-10046,
  author       = {Jia Li and
                  Xianjie Shi and
                  Kechi Zhang and
                  Lei Li and
                  Ge Li and
                  Zhengwei Tao and
                  Jia Li and
                  Fang Liu and
                  Chongyang Tao and
                  Zhi Jin},
  title        = {CodeRAG: Supportive Code Retrieval on Bigraph for Real-World Code
                  Generation},
  journal      = {CoRR},
  volume       = {abs/2504.10046},
  year         = {2025},
  url          = {https://doi.org/10.48550/arXiv.2504.10046},
  doi          = {10.48550/ARXIV.2504.10046},
  eprinttype    = {arXiv},
  eprint       = {2504.10046},
  timestamp    = {Thu, 25 Sep 2025 08:48:22 +0200},
  biburl       = {https://dblp.org/rec/journals/corr/abs-2504-10046.bib},
  bibsource    = {dblp computer science bibliography, https://dblp.org}
}

4. The empirical results look disappointing. For example, from Table 1, it seems that Gemini 1.5-Pro with DDG can already achieve 0.587 for AGA-Code, but the performance of GraphRAG-SCG (which is on top of Gemini 1.5-Pro and with multiple types of graph information) in Table 2 is much lower than that.
5. This paper claims to target smart contract code generation, but GraphRAG-SCG does not seem to be specific to smart contract code generation. That is to say, it is unclear how the proposed technique actually considers specific characteristics of smart contract programs.

**Questions:**

1. In GraphRAG-SCG, how do you retrieve the graph? Is there a graph database to retrieve against? If so, how to construct the database?
2. Can you confirm that GraphRAG-SCG is actually less effective than its base model (i.e., Gemini 1.5-Pro)?
3. Is GraphRAG-SCG specific to smart contract code generation? Can GraphRAG-SCG be used for general code generation?

---

### Official Review · Reviewer_HSL3 · 2025-11-01

**Soundness:** 2
**Presentation:** 2
**Contribution:** 2
**Rating:** 4
**Confidence:** 5

**Summary:**

This paper introduces GraphRAG-SCG, a framework that integrates graph representations—specifically function-call graphs (FCGs), data-dependency graphs (DDGs), and business-constraint facts—with large language models (LLMs) for smart contract code generation. The authors propose a dual-layer "semantic-control" graph index to retrieve structurally relevant subgraphs, which are then linearized and injected into LLM prompts. Experiments on 40,000 real-world smart-contract pairs demonstrate improvements over baselines in metrics such as RAGA-Code, CodeBERTScore, and CodeBLEU.

**Strengths:**

1.  The research addresses a novel and timely problem by combining graph-based program representations with retrieval-augmented generation for smart contract code synthesis, a direction not widely explored in existing literature.
2.  The proposed method shows measurable improvements in multiple automated evaluation metrics, suggesting that the integration of structural cues can enhance the quality of generated code.

**Weaknesses:**

1.  The claimed contributions are incremental. The idea of augmenting LLMs with structural information, such as FCGs and DDGs, has been explored in broader code generation contexts, and the adaptation to smart contracts does not constitute a significant conceptual advance.
2.  The experimental design is insufficient. The baselines are limited, and there is no case study or qualitative analysis to demonstrate the practical superiority or failure modes of the generated contracts.
3.  The writing and presentation suffer from several issues, including incomplete references, misformatted citations, unclear figures, and awkward pseudo-code layout, which hinder readability and reproducibility.

**Questions:**

1.  In the RELATED WORK section, there is very little discussion of research on smart contract code generation. Is this because the field is relatively understudied, or have many relevant works been overlooked?
2.  At line 143, there appears to be a citation formatting error.
3.  Figure 1 appears to be very unclear.
4.  The pseudo-code layout from lines 335 to 349 looks odd.
5.  The experiments are quite insufficient. For example, there are too few baseline models, and no case analysis is presented.

---

### Official Review · Reviewer_fYqS · 2025-11-02

**Soundness:** 2
**Presentation:** 2
**Contribution:** 2
**Rating:** 2
**Confidence:** 3

**Summary:**

This paper proposes GraphRAG-SCG, a retrieval-augmented generation framework for automated smart contract code generation. The approach constructs a dual-layer "semantic-control" graph index that integrates function-call graphs (FCGs), data-dependency graphs (DDGs), and business constraints to guide LLMs during code generation. Through graph traversal and embedding-based retrieval, relevant subgraphs are identified and injected into prompts. The authors evaluate on 40,000 smart contracts from Ethereum, demonstrating improvements of 13.1%, 5.8%, and 2.4% in RAGA-Code, CodeBERTScore, and CodeBLEU metrics over baselines.

**Strengths:**

+ Novel Integration: The paper presents an innovative combination of graph-based program analysis with RAG for code generation, addressing limitations of purely textual retrieval approaches.
+ Comprehensive Graph Representations: The use of multiple graph types (FCGs, DDGs) plus symbolic constraints provides rich structural context that captures both control-flow and data-flow semantics.
+ Multiple Evaluation Metrics: Uses three complementary metrics (AGA-Code, CodeBERTScore, CodeBLEU) to assess different aspects of code quality.

**Weaknesses:**

Unfair Experimental Comparison: Only comparing few-shot GraphRAG-SCG against fully fine-tuned models CodeT5+, CodeBERT is not appropriate. The former is empowered by the latest most powerful LLMs, whose parameters are much more than small models like CodeT5+ and CodeBERT. Thus GraphRAG-SCG should also be compared using prompt-based methods, e.g., “SolEval: Benchmarking Large Language Models for Repository-level Solidity Code Generation”. Additionally, the methods in baselines are out-of-date, some new methods could also be considered, such as “Guiding LLM-based Smart Contract Generation with Finite State Machine”.

Missing Critical Implementation Details: Algorithm 1 & 2 are high-level pseudocode without sufficient implementation details.
- No description about the input and output of ANN, and the details of selecting node candidates.
- No explanation about the formula ‘struct(Sv)’.
- No figure to illustrate how the final prompt looks like.

Missing Experiment Settings: To prove the effectiveness of subgraph scoring and selection, the following experiments should be added.
- Combinations of various values of the hyperparameters ‘alpha’ and ‘beta’.
- Randomly select the subgraphs without considering the score.

Presentation
- In Line143, a reference error.
- Many unclear texts and symbols occur in Figure 1.
- In Line 485, miss the punctuation mark at the end.

**Questions:**

1. How effective is GraphRAG-SCG when compared with other prompt engineering-based methods?
2. How do hyperparameters ‘alpha’ and ‘beta’ affect the effectiveness of the method?
3. Are there test suites for each data point in the benchmark? The superior pass rate in test cases may be more persuasive.
4. Can you provide the prompts used?

**Details Of Ethics Concerns:**

N.A.

---

### Meta-Review · Area_Chair_qM3o · 2026-01-04

**Summary:**

While the idea of incorporating graph structure into RAG for code generation is timely and potentially impactful, reviewers raised substantial concerns regarding experimental fairness, baseline strength, implementation clarity, novelty, and empirical validation, which ultimately outweigh the reported metric improvements.

**Reviewer Concerns:**

Unfair and weak experimental comparisons.
Incremental novelty relative to prior work.
Limited analysis and ablation.
Evaluation metrics and task validation.

**Reviewer Scores:**

There is no author reply, therefore the negative reviews will remain.

---

### Decision · Program_Chairs · 2026-01-26

Reject